# Simulation Analysis of Bus Passenger Boarding and Alighting Behavior Based on Cellular Automata

Yunqiang Xue [1,2,*], Meng Zhong [1], Luowei Xue [3], Bing Zhang [1], Haokai Tu [1], Caifeng Tan [1], Qifang Kong [1] and Hongzhi Guan [1,4]

1  School of Transportation Engineering, East China Jiaotong University, Nanchang 330013, China; zhongmeng@ecjtu.edu.cn (M.Z.); zhangbing@ecjtu.edu.cn (B.Z.); tuhaokai@ecjtu.edu.cn (H.T.); tancaifeng@ecjtu.edu.cn (C.T.); kongqifang@ecjtu.edu.cn (Q.K.); hguan@bjut.edu.cn (H.G.)
2  School of Transportation, Southeast University, Nanjing 210096, China
3  Jiangxi Provincial Institute of Transportation Science, Nanchang 330200, China; luoweixue@ecjtu.edu.cn
4  College of Architecture and Civil Engineering, Beijing University of Technology, Beijing 100124, China
*  Correspondence: xueyunqiang@ecjtu.edu.cn; Tel.: +86-132-4173-2469

**Abstract:** Bus passengers' boarding and alighting behavior is important content when researching bus operation efficiency. This paper uses an improved cellular automata (CA) model and introduces four dynamic parameters to study individual behavioral characteristics of bus passengers' boarding and alighting behavior. The research on the relationship between the macro pedestrian flow formed by the interaction between the individual passengers and the stop time of the bus station was realized. Then it was modeled for different situations, and the general update rules of CA were set based on realistic situations. The passenger boarding and alighting behaviors of the No. 245 bus route in Nanchang, China were simulated, and the simulation results of four different door layouts and passenger boarding and alighting modes were compared. It was found that when the passenger loading rate in the bus reaches 65%, the passenger boarding rate has an obvious tendency to slow down; the width of the door has a direct relationship with the passenger alighting efficiency, and the bus stopping time can be reduced by adjusting the width of the alighting door; a strategy which allows passengers board on the bus via the alighting door may effectively reduce the bus stopping time when there are many passengers boarding on the bus. Using strategy four, simulation research found that Bus No. 245 can reduce the stopping time by 40–50% in some station scenarios. Research results show that the CA model has certain practical value and can provide a theoretical reference for public transportation control and management.

**Keywords:** pedestrian traffic flow; cellular automata; dynamic parametric model; bus load factor; stopping time

## 1. Introduction

In recent years, with the growth of urban population and the increase in car ownership, urban traffic congestion has become an increasingly serious problem, and has become an important factor restricting urban development. The development of public transport is an important way to improve the utilization of traffic resources and alleviate traffic congestion, and has a significance for sustainable transportation. Due to the high passenger capacity of buses, less occupancy of road resources, low energy consumption, relative environmental protection and low transportation cost, more and more cities pay attention to the sustainable development of public transportation, so that public transportation is organically integrated with urban construction and development [1].

Urban conventional buses are the main body of public transportation, and their service level greatly affects the travel choices of residents. Improving the level of bus service can reduce the unit cost of bus travel time, which in turn enhances the attractiveness of public transport [2]. The operation efficiency of buses is one of the manifestations of the level of bus

service and is influenced by the stopping time at stations. When the platform stopping time is long, the resulting delays may lead to bus bunching problems, which have a significant impact on bus punctuality and operational negative efficiency [3]. This is not conducive to the sustainable development of public transportation. Bus stops, as nodes that connect bus users to bus vehicles, undertake the travel of passengers. At the same time, the delay of getting on and off at bus stops often becomes a bottleneck in the urban transportation network system. Therefore, studying the boarding and alighting behavior of passengers can improve the bus punctuality rate, avoid the bus bunching phenomenon, and at the same time play an important role in optimizing the transportation network system.

The behavior of passengers boarding and alighting falls within the realm of pedestrian flow research. Studying the microscopic individual behaviors of passengers getting on and off at the station can realize the characteristics of the macro pedestrian flow formed by the interaction between individuals, and obtain the stopping time of the bus at the station. The study of pedestrian traffic flow can be traced back to the exploration of pedestrian traffic flow by Predtechenskii and Milinski of the former Soviet Union's Architectural Research Institute (VAKH) in 1937 [4]. In the late 1950s, the theory of pedestrian traffic flow began to be studied systematically. Although there is a relatively mature theoretical basis for the study of pedestrian traffic flow, there is less research on passenger boarding and alighting time. In particular, there is a gap in the study of the impact of passenger boarding and alighting on bus operations in terms of bus bunching. Most of the research on bus passenger boarding and alighting behavior has focused on its characteristics, as well as the inference and discrimination of bus stops. It is necessary to carry out a simulation study of bus passenger boarding and alighting behavior. It can discover the impact on the reliability of bus operations and provide theoretical support for subsequent related studies.

Pedestrian traffic simulation research can be divided into two levels: macro and micro [4]. At the macro level, the research is mainly conducted to determine which activity events occur, and only considers the impact of external factors on pedestrian simulation without considering the interaction between pedestrians, as well as pedestrians and facilities. The micro-level mainly analyzes the behavior of a single pedestrian, determines the decision-making behavior of a single pedestrian at the next moment, and considers the movement behavior of the pedestrian, the interactions among the pedestrians, facilities and other traffic. In summary, the analysis of bus passengers' getting on and off behavior is more in line with the microscopic research. Therefore, the micro simulation modeling research on the pedestrian flow of bus passengers is carried out in this research.

Among the studies on micro models of pedestrian flow simulation, the representative ones are the agent-based model [5,6], social force model [7], CA model [8–10] and lattice gas model [11]. Among them, the social force model belongs to the continuous model, and the others are discrete models. At present, pedestrian simulation models based on multi-agents are generally proposed based on the belief–desire–intention (BDI) model; by assigning characteristics to different groups of people and formulating rules, passengers in the bus can be simply simulated [12]. However, the agent is only a modeling and programming method, and the average modeler is not sure what to do and how to achieve it, so it is difficult to master [13]. The social force model is a microscopic simulation model proposed by Helbing et al. [14] based on the molecular dynamics model to simulate pedestrian flow. The model proposes that the movement of pedestrians is jointly determined by three forces, which are: self-driving force due to the attraction of pedestrians by the destination; interaction force between pedestrians and pedestrians; interaction force between pedestrians and the surrounding environment. It can realistically describe pedestrian dynamic behavior [6], and has many applications in the simulation of passenger behavior inside buses and subways [15–17]. Although the social force model can better describe the movement process and behavior characteristics of pedestrians in different scenarios and different situations, the model has disadvantages such as high computational complexity, high resource consumption, and long calculation time in the process of simulation realization. The lattice gas model is one of the discrete models developed from the CA approach, and

was first proposed by Muramatsu et al. [18,19], and the model has been used in simulations to study the characteristics of pedestrian flow in several different situations, including unidirectional flow, counter flow, and cross flow. It is usually combined with other methods and is mostly used for pedestrian evacuation and considering pedestrian behavioral preferences [20]. When simulating evacuation, the lattice gas model pays more attention to the physical fluid characteristics shown in group evacuation. The CA model has the advantages of fewer parameters, simple structure, convenient computer modeling, and fast running speed in describing the internal laws of complex systems. This paper conducts simulation analysis on passengers getting on and off the bus and the microscopic individual behaviors of passengers in the bus. The CA based on dynamic parameter model proposed by Hao Yue has guiding significance for micro pedestrian flow simulation [21–23]. Some of the dynamic parameters cited in this article refer to Yue Hao's chapter content (Simulation Study of Lateral Interference Pedestrian Traffic Flow), such as direction parameters and forward parameters.

The CA model is a grid dynamic model in which time, space, and state are discrete, and the spatial interaction and time causality are local. It has the ability to simulate the spatiotemporal evolution process of complex systems. Blue et al. [24] proposed that pedestrian flow is inherently more complex than vehicle flow, and the development of a microscopic model of pedestrian flow has always been a difficult task for researchers. In recent years, the CA model has been continuously developed, which can be divided into two aspects: the study of typical phenomena of pedestrian traffic in non-emergency situations and the study of pedestrian evacuation behavior in emergency situations. In non-emergency situations, Zhang, Nowak, Feliciani et al. [25–27] studied the characteristics and manifestations of pedestrian flows in specific environments. In emergency situations, the CA model is mainly the study of dense crowd events. Gwizdałła [28] investigated the influencing factors of the pedestrian evacuation process. Pan S, Zhou J. et al. [29,30] studied the evacuation process of pedestrians in different scenes such as loop floors, subway halls, classrooms, and teaching buildings. The CA model has been continuously improved in simulating pedestrians, and the simulation results are in good agreement with reality [26,31,32]. It is widely used in the analysis of dense crowd events such as pedestrian travel on the road, bus passenger boarding and alighting, and fire crowd evacuation [8,33,34].

In different areas and situations inside the bus, the error of social force model processing cannot be ignored. Its use for passenger behavior simulation has certain limitations. The intelligent body model requires a combination of modeling and programming methods, and is difficult for non-specialists to control and grasp when simulating passengers inside the bus. The CA model and the lattice gas model are used as discrete models, both of which have high computational efficiency, but the lattice gas model pays more attention to the physical fluid characteristics shown in the crowd evacuation when simulating evacuation. Compared with others, the CA has a greater advantage in simulating crowded pedestrian flows in complex situations due to the simplicity of model computation, variable update rules and high flexibility in pedestrian flow simulation.

The CA can handle these complex traffic behaviors by changing the cell update rules, among which the dynamic parameter model has advantages in dealing with the crowded state. Therefore, the dynamic parameter model is chosen to simulate and analyze the boarding and alighting behavior of passengers in this paper. This study aims to reduce bus stopping time by considering the impact of passenger boarding and alighting behavior, full occupancy and door width on bus stopping time from a microscopic perspective. Before constructing a passenger boarding and alighting model, the behavioral characteristics of the passenger boarding and alighting process need to be investigated and analyzed. According to Wang Jianmei (2015) [35] and Zhang Ruochen (2017) [15], who investigated and analyzed the behavioral process of regular bus passengers boarding and alighting in Chengdu and Nanjing respectively, it was found that the characteristics of the bus passenger boarding and alighting process were similar between these cities. At the same time, the study referred to the survey methods of these two pieces of literature and conducted

a video recording survey of passengers' boarding and alighting processes by following and stopping the bus. Based on the fieldwork, it was found that the case cities and the literature had similar modes of conventional bus operation, consistent boarding and alighting rules, similar characteristics of passenger boarding and alighting behavior, and slightly different characteristics of passenger boarding and alighting times. This model reveals the impact of bus full load rate on passenger boarding and alighting rates from a microscopic perspective. It is also applicable to other examples, and only needs to be modified in two aspects: one is to set corresponding bus cell spaces for different vehicles, and secondly is to update the rules of cells according to specific cases to make changes to the corresponding part of the rules.

The CA model has a greater advantage in simulating pedestrian flow. However, when considering the congested state, the CA will be biased, especially when there is a high density of pedestrians; the pedestrian flow will be stagnant due to congestion, which is difficult for the general CA to simulate. In real life, pedestrians will be able to pass in congested state by means of communication and communication so that pedestrians do not stagnate. To solve such problems of CA, the rest of this paper proceeds as follows: Section 2 develops the CA model for bus passenger boarding and alighting. We first give an overview of the bus passenger boarding and alighting model and define the parameters. Then we give the calculation method of these parameters, and then develop the CA update rules for bus passengers according to the actual situation for the model. In Section 3 of the simulation analysis, this paper obtains the relationship between the number of passengers in the bus, the number of passengers getting on and off the bus, and the bus stopping time by simulating the passenger pedestrian flow of bus No. 245 in Nanchang. Considering that the door width has an effect on passenger boarding and alighting [35], this paper compares the station stopping time of four boarding and alighting strategies with different door widths in the case of 20 boarding passengers and 10 alighting passengers, providing theoretical support for the later control research and management of bus lines. Next, Section 4 discusses the results. This study concludes with a discussion of the managerial and theoretical implications, suggesting limitations of the study and avenues for future research. At present, there are few researches on bus passengers based on micro-simulation, so this article has a breakthrough development significance in the field of bus passenger research.

## 2. CA Model for Bus Passenger Boarding and Alighting

The bus passenger boarding and alighting model proposed in this paper divides the bus interior space into a number of positive hexagonal cells, the size of which is determined based on the space required by the actual passengers riding the bus. However, it should be noted that the cell size is different from the actual size of the space that is occupied; the cell size is the safe space required by the passenger in the current situation. Therefore, the cell is larger than the actual space occupied by the passenger. Each cell corresponds to different parameters according to the internal layout and location of the bus. The simulation process of passenger passage inside the bus is also discretized into equal simulation *steps*, and within each discrete simulation *step*, the passing passenger can choose to move with unit speed $Vs = 1$ *cell/step* or stay in the current cell. Each cell has a corresponding connected cell (adjacent cell), and the cell chosen by the pedestrian within the cell per simulation step must be either the current cell or an adjacent cell.

As a practical matter, all passengers are divided into three categories: boarding passengers, alighting passengers, and stranded passengers. Boarding passengers are passengers whose destination cell is the standing area or seat in the bus and who have not reached the target cell. Alighting passengers are passengers whose destination is to get off through the door and have completed their journey. Stranded passengers means passengers who have finished boarding and have not reached the target stop. These three categories of passengers have different purposes within the bus, with correspondingly different parameters and behaviors, and they are switchable between the three categories. For example, when

a seat appears near the stranded passenger, the seat selection behavior of the stranded passenger makes him/her an on-board passenger; when the stranded passenger reaches the destination stop, he/she becomes an alighting passenger.

For the effective space inside the bus, different parameters are set according to the characteristics such as boarding and alighting attributes, standing position, and seating. Therefore, this research model combines different bus layouts, characteristics of the cells, and characteristics of the passengers inside the cells and sets up cell update rules.

### 2.1. Dividing Principle of Bus Interior Layout

Each seat is used as a cell. For the standing area, the space occupied by an average passenger is divided as a cell when stationary. Considering the accessibility between cells, the principle of cell division is shown in Figure 1. For the cell $x$ in the figure, denote its neighboring cells as $C_{x,i}$, where $i$ is all neighboring labels from the counterclockwise count of the neighboring cells directly below cell $x$, with a minimum value of 1 and a maximum of 6, and each $C_{x,i}$ belonging to a cell in the panel of bus cells (see Figure 1a).

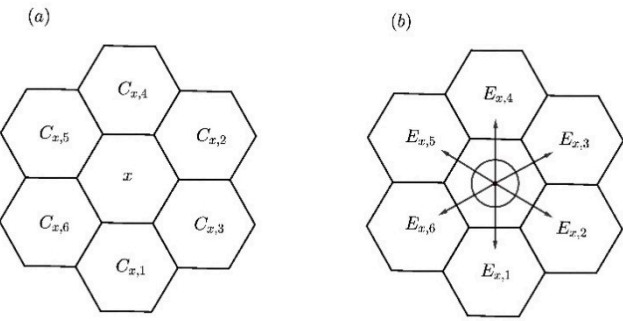

**Figure 1.** Adjacent cells (**a**) and the corresponding direction parameters (**b**).

### 2.2. Dynamic Parameter Setting

In this paper, some new dynamic parameters are proposed to adapt to different practical situations inside the bus: a potential-based direction parameter, an attractiveness parameter, a cell-occupancy parameter, and a forward parameter.

To compensate the problem of unbalanced gain acquisition by the dynamic parameter model, this study uses a hexagonal CA. For each time step, the hexagonal CA is the same; the pedestrian walks from any direction towards the next cell, the walking distance is the positive hexagonal side length and the walking time is the same, avoiding the situation that the square cell in the dynamic parameter model has different distances from the centers of the eight adjacent cells. Because of the fixed unit simulation step length of the CA, the square hexagonal cell can more accurately represent the actual phenomenon in building the pedestrian flow CA model compared to the square cell, which cannot aptly represent the direction parameter of the adjacent cells.

#### 2.2.1. Direction Parameter

The direction parameter $E_{x,y}$ refers to the potential energy difference of the cell from the target cell. It reflects the direct movement benefits that pedestrians can obtain by moving one time step under the condition of judging the traffic conditions in their own moving area. Its simplified form is the following formula:

$$E_{x,y} = p_x - p_y \tag{1}$$

where: $y$ denotes the neighboring cell of $x$, i.e., $y = C_{x,i}$ and $p$ is the potential energy relative to the target cell of $x$. Potential energy is similar to potential energy in mechanics. In mechanics, potential energy is the energy possessed by an object or system due to its position or potential shape. Similarly, the potential energy of a cell proposed in this paper is based on a certain cell as the base point and the energy possessed by other cells. Generally,

each cell takes its own target cell as the origin of potential energy, and the energy at other positions is the potential energy relative to this origin.

For passengers, the maximum travel distance per unit time *step* is 1, so each passenger can choose the neighboring six cells as the direction of travel for the next time step or stay in the current cell, and the location gain of the action is the potential energy difference between cells, i.e., the direction parameter (see Figure 1b).

For the direction parameter, the potential energy of each boarding passenger at the target position $d$ is 0, and the potential energy of $x \in N_d$ its neighboring cells is 1. $N_d$ is the set of cells adjacent to cell $d$, and so on, labeled $p_{x_1} = p_{x_0} + 1$, where $x_0$ is the cell that has been labeled with the potential energy, $x_1 \in N_{x_0}$, and in the same way, the potential energy of the disembarking passenger is calculated with the following car cell $a$ as the target cell.

Assuming that the objective of the passage cell is $d$, then for the passage cell $c$ the potential energy is calculated as follows.

A. Mark the potential energy of $p_{c,d} = 0$ target cell $d$.
B. Divide all cells into labeled potential cells $M$ and unlabeled cells $N$.
C. Calculation $p_m = \max(x), x \in M$.
D. Mark $p_c = p_m + 1$, where $x \in N \cap C_y, y \in M$.
E. If $N = \varnothing$, end; otherwise, return to step 2.

The direction parameter $E_{x,y} = p_{x,x} - p_{x,y}$ of the cell $x$ to $y = C_{x,i}$ is obtained by calculating the potential energy difference, and $E \in \{-1, 0, 1\}$.

For a passenger of cell $x$, its direction parameter from cell $i$ to cell $j$ is noted as $E_x(i, j)$

$$E_x(i, j) = p_{x,i} - p_{x,j} \tag{2}$$

### 2.2.2. Attractiveness Parameter

The attractiveness parameter is mainly the expression of the attractiveness of the facility or area to passengers. For practical purposes, seats on buses are more attractive to passengers than standing space, and boarding passengers prefer a seat as a target area (lagging area). When the seats are full, other areas are selected as lagging areas. Bus passengers mostly use the comfort level of the lagging area as a basis for judging whether to choose that area or not. Generally, stranded passengers move when there are free seats.

Based on the above-mentioned actual situation, this model proposes the premise hypothesis that stranded passengers move and go to a more attractive free cell when it appears and that the cell only accepts the closest passenger to itself. In this paper, only the effect of seats and standing areas on passenger attractiveness is considered, so for the model attractiveness parameter in this paper, zero represents standing areas; one represents seats.

### 2.2.3. Cell-Occupancy Parameter

The cell-occupancy parameter, improved on the empty parameter of the original dynamic parameter model, is adapted to the presence of stranded passengers in a congested state, and to the dynamic parameter adopted to consider pedestrian passage. This parameter represents the state of the cell, and there are three basic states: empty cell, stranded cell, and passing cell (see Equation (3) for details). An idle cell refers to a cell in which no passenger is present. A stranded cell refers to a cell that does not get off at a bus stop and does not move and in which only one stranded passenger is present. A passing cell refers to a cell in which one boarding or alighting passenger is present and in which one stranded passenger may be present.

The values of the occupancy dynamic parameter matrix elements are the following.

$$O_i = \begin{cases} 0 & \text{Empty cells location} \\ 1 & \text{Retained Cells} \\ 2 & \text{A cell occupied by passenger getting off} \\ 3 & \text{A cell occupied by a boarding passenger} \\ 4 & \text{Interaction cells occupied by boarding passengers} \\ 5 & \text{Interaction cells occupied by getting off passengers} \end{cases} \quad (3)$$

These are the cells that have already reached the target location in the vehicle. These cells usually do not change their location easily unless they get off the vehicle or a seat becomes available nearby.

### 2.2.4. Forward Parameter

Passengers tend to move towards an open position in front of them as they move, so that they will have more choice of locations as they move, so as to avoid crowding and passenger interaction. Therefore, the forward parameter uses the fact that the passenger's next target position is not occupied by the cell to reflect the attraction degree of the target position to the passenger. The forward parameter refers to the number of six cells adjacent to the cell for which the direction parameter is positive.

The forward dynamic parameter matrix values are:

$$Fwd_{i,j} = |\{x|E_\alpha(i,j) > 0\}| \quad (4)$$

In the formula: $Fwd_{i,j}$ is the forward dynamic parameter of the passenger from cell $i$ to cell $j$.

In addition to the direction parameters, attractiveness parameter, occupancy parameters and forward parameters, each cell has characteristic parameters: lag time, target cell, cell function, and adjacency parameters. Lag time $t_i$ to the simulation step that the boarding and disembarking cells stays in the current cell and is used to control the travel speed of passengers in different situations. The target-cell parameter refers to the destination of the boarding and alighting cells.

The model divides the cells into five categories according to the cell-function parameter $B$: boarding cells $B_1$ (the upper door area where passengers purchase tickets to board the bus), alighting cells $B_2$ (the exit where passengers leave the bus cell space), standing cells $B_3$ (the standing area inside the bus), seat cells $B_4$ (the seats inside the bus), and correction cells $B_5$. The correction cells are $B_5$ used to adjust the parameters of the CA during the simulation to achieve realistic results (see Figure 2), and the shaded cells 5 and 42 in the figure are the correction cells, which are used to ensure that the boarding and alighting service times match the actual ones. The gray area in Figure 2 is the seat cell.

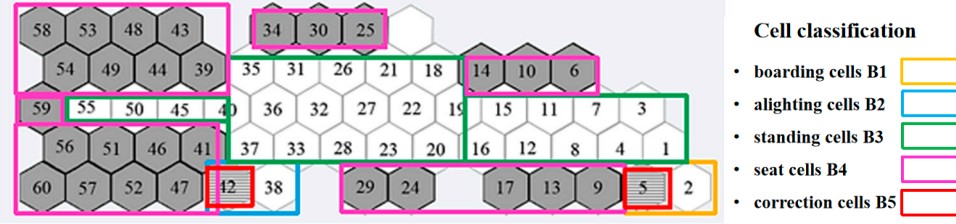

**Figure 2.** Bus cell space.

### 2.3. Model Evolution Rules

2.3.1. Crowded Passing Rules

Usually there is a certain safety distance between passengers and passengers. When a passenger gets on and off the bus, but there are more stranded passengers and they cannot directly pass through the aisle to reach the destination area, that passenger will request the stranded passenger to give a passage. At this time, the passenger who is getting off cannot move forward at normal speed, and every time they pass the crowded area they will interact with the stranded passenger in order to move forward. Thus, there will be a trend of travel in Figure 3. The hollow circles in Figure 3 indicate the stranded passengers, the solid circles indicate the route of the alighting and disembarking passengers through the area, and the arrows indicate the direction of travel. To implement the rule in a CA, this paper simplifies the interaction process: each lagging cell a passenger passes through interacts directly with the lagging cell, and the cell adjacent to the lagging cell determines whether to accept other passing passengers by determining whether to engage in the interaction.

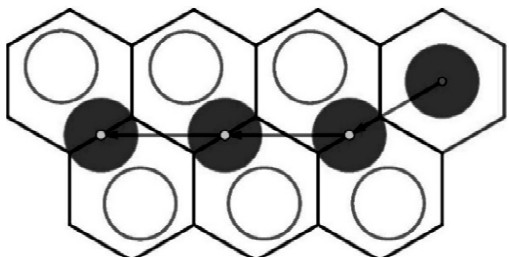

**Figure 3.** Diagram of the traffic model in a congested state.

Figure 4 represents a schematic diagram of an on/off passenger interacting with a lagging passenger in order to pass through a crowded region. The solid black circles represent the passing passenger, and the passing target is the vacant upper-side cell. The passenger in Figure 4a prepares to interact with the right-side lagging passenger and then shares a cell region with the right-side lagging passenger (Figure 4b). We consider that several cells below this interaction cell are affected, and these cells are relatively far from the target cell of the passing passenger. At this point, the left-lagging passenger is not engaged in the interaction. However, when the passing passenger approaches the target cell, because the interaction cell is close to the upper side cell, the left-lagging passenger is affected at this point and the passing passenger needs to interact with that lagging passenger, which is not explicitly expressed in the figure. Usually a stranded passenger will only interact with one passenger at a time, so both of these cells no longer receive interactions from other passengers. When the interaction is complete (Figure 4c), the passenger arrives at the target cell (Figure 4d) and becomes a new lagging passenger, and that cell becomes a new lagging cell.

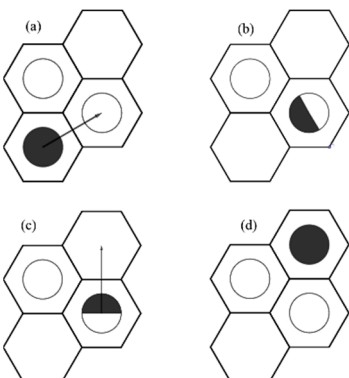

**Figure 4.** Schematic diagram of cell interaction.

### 2.3.2. Movement Benefit Rules

The seven candidate locations in the passenger mobility domain have their own dynamic parameter and movement benefit. The dynamic parameter and transition payoff vary with the type of passenger in the center of the mobility field, the occupancy of the cells in the mobility field, and the passenger's field of view. The formula for calculating movement benefit $p_{i,j}$ is:

$$P_{i,j} = \begin{cases} T_{i,j} + Fwd_{i,j} & O_j \leq 1 \\ -M & O_j \leq 1 \end{cases} \tag{5}$$

of which

$$T_{i,j} = \begin{cases} 0 & O_j = 1, E_{i,j} > 0 \\ 3 & O_j = 0, E_{i,j} = 0 \\ 6 & O_j = 0, E_{i,j} > 0 \end{cases} \tag{6}$$

In the formula, $P_{i,j}$ is the passenger's moving revenue from cell $i$ to cell $j$; $T_{i,j}$ is the weighted value of passenger from cell $i$ to cell $j$; M is an infinite value.

As an example of movement benefits, the passenger distribution is assumed as shown in Figure 5.

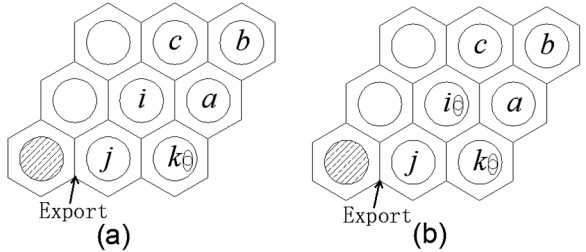

Export     Export
(a)        (b)

**Figure 5.** Schematic diagram of movement benefit calculation example. (Note: (**a**) Schematic diagram of passenger movement benefit calculation example, in which $k$ has stranded cells and the others are empty. (**b**) Schematic diagram of passenger movement benefit calculation example, in which $i$ and $k$ also have stranded cells, and the others are empty).

The movement benefits selected by passengers are divided into priorities, and the priorities are divided into 0 to 8 levels according to different situations. ① When the direction parameter $E < 0$, the cell movement has no benefit, and it is not considered at this time; that is, the cell has been occupied by passengers getting on and off, and when it is unavailable, $P$ is an infinite negative value. ② The value of $Fwd$ is 0 to 2 (integer). In Figure 5a, if $i$ has no passengers, $Fwd_{b,a} = 1$; in Figure 5b, if $i$ has passengers, $Fwd_{b,a} = 0$.

Then in Figure 5a, the action benefit of the passenger moving from $b$ to $j$ is calculated as follows:

$$\begin{aligned} P_{b,a} &= 6 + 1 = 7 \quad P_{a,i} = 6 + 2 = 8 \\ P_{a,c} &= 3 + 2 = 5 \quad P_{a,k} = 0 + 1 = 1 \\ P_{k,j} &= 6 + 0 = 6 \quad P_{k,i} = 3 + 1 = 4 \\ P_{i,j} &= 6 + 0 = 6 \quad P_{i,n} = 3 + 1 = 4 \\ P_{i,k} &= -M + 1 = -(M-1) \end{aligned} \tag{7}$$

In Figure 5b, $P_{b,a} = 6 + 0 = 6$.

Interaction Judgment.

According to the action step and potential energy, if a stranded cell c, it will walk t steps to reach the next cell (target cell), when there is an adjacent cell with a potential energy greater than 0 At the same time, when the step length to the adjacent cell is less than the step length to the target cell, then the target cell is an interactive cell, and this action process is also called interactive behavior. That is, assume that each interaction takes $t$ *steps*, and note that $t_m = [t/2]$ for $c \in \{x | O_x = 1\}, \exists y$, such that $y \in C_c, E_y(c,y) > 0, t_y < t_m$, then $c$ is the interaction cell.

Judging from the cell dynamic parameter value, it can be seen from the dynamic parameter matrix element formula that when the cell dynamic parameter value is greater than 3, the cell belongs to the interactive cell, and the process that targets this type of cell belongs to the interaction. That is, for a cell $c \in \{x|O_x \neq 1\}$, if $c \in \{x|O_x > 3\}$, then $c$ is an interacting cell.

### 2.3.3. Cell Update Rules

According to the congestion passage model mentioned earlier, the maximum number of passengers accommodated per cell is two. According to field surveys on public transport, the cell space of this model for the application to buses has the following rules:

A. The maximum velocity of each passenger in the model is $V_{\max} = 1/step$, i.e., the boarding and alighting passengers can only move one cell length in each time step $t$. Pedestrians have seven optional locations (including its own location) as their next target location (See Figure 1).

B. In a time step $t$, the passenger's cell choice is determined based on the gain $p_i$ obtained from the action to the six adjacent cells. When there is no congestion in the passenger's direction of travel, the passenger will choose the cell with the largest and most positive gain as the target location for the next step.

C. The stranded passenger moves only in the state when the more comfortable cell appears to be free; conversely, if a more comfortable cell (seat) appears, the cell properties of the stranded passenger can be transformed into the boarding cell and the comfortable cell is selected as the target cell.

D. When a crowded condition occurs, the passing passengers and stranded passengers conduct an interaction that allows the stranded cells to make room for passage. For each passing unit length, it is necessary to interact with and pass through more than two resident cells. Each cell containing a lagging passenger can only receive one interaction from a passing passenger or receive one pass occupancy at the same time. Cells occupied by a single row of seats do not accept interactions.

E. When a congested aisle shows no displacement of passing passengers within a unit of time, the aisle stops accepting other passengers.

F. The duration of passage of a passing passenger in a cell is related to the state of the cell the passenger is in and the next target cell.

G. When it appears that a cell is taken as the next step by more than one passenger, the cell receives a random passenger and the other passengers recalculate the parameters and choose to update the cell.

### 3. Simulation Analysis

Simplify the congestion passage model due to the simplicity of the CA itself: first, create a panel of cells according to the layout of the bus. Each cell represents the area that passengers can reach, and the panel parameters are: the neighboring cell $C$, which indicates the connectivity between different cells; and the functional parameter $B$.

Assumptions:

- The characteristics of each category of passengers are the same.
- Passengers will start preparing to get off the vehicle and proceed to the getting off area when they are prompted to do so.
- When the bus stops, the boarding passengers have all reached the upper doors of the bus and begin to board, while the alighting area begins to alight.

### 3.1. Case Scenarios

This paper simulates Bus No. 245 in Nanchang, which has a rated capacity of 80 passengers. Bus No. 245 has an effective width of 70 cm at the front door and 110 cm at the rear door. When there are fewer passengers in the bus, the safety control for passengers is larger, and the size of the metric automaton panel used at this time is 61 (see Figure 2), which can accommodate up to 56 passengers, corresponding to a full bus occupancy rate of 70%.

When the passenger fullness rate reaches 70%, the panel is replaced and the parameters are adjusted. The bus is initialized using the CA model, and the space inside the bus is divided into 61 cells. The cells are divided into five categories according to their functions: boarding cells $B_1$, alighting cells $B_2$, standing cells $B_3$, seat cells $B_4$, and correction cells $B_5$. The bus route is a two-way line from the north exit of Wenjiao Road to Honggutan New City. The whole site is 15 km and passes through 24 stops. The bus routes and numbers are shown in Figure 6. According to the investigation of car following and stationing in the first half of 2019, the average travel time of the bus during peak period is 40 min, and during peak period it is 50–70 min. The peak hour traffic of this line is about 2000 pax/h.

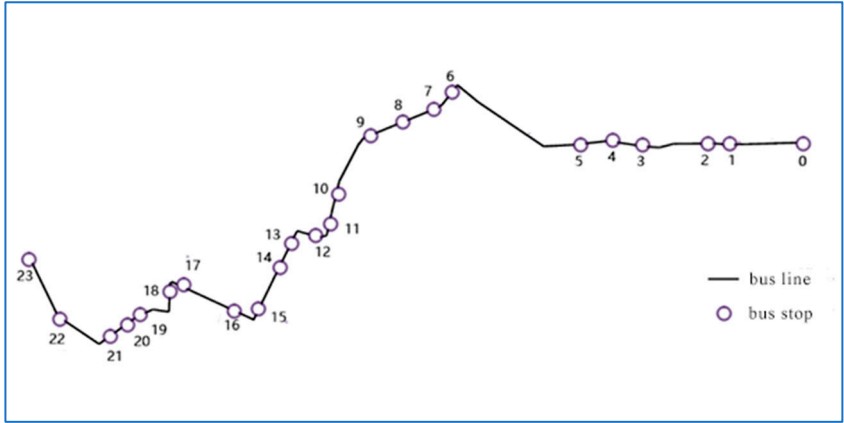

**Figure 6.** No.245 bus route and station number diagram.

According to the literature [15,35] survey analysis of the passenger boarding and alighting process, with reference to passenger boarding and alighting behavior and time characteristics, combined with the on-site bus station survey and bus follow-up survey, we know that the average ticketing time for boarding passengers is about 2 s/person, the free flow of passengers traveling in the bus is about 1 m/s, and the space occupied per passenger at 75% occupancy is about 0.2 m$^2$. The speed decreases to 0.25 m/s as the passenger passes through the crowded area and decreases when normal passage is disturbed. Without the correction cell, the established cell space in the boarding cell connects to only one standing cell, making the cell determined to be blocked and the service frequency less than 2 *s/person*, and the correction cell 5 in Figure 2 does not accept any passenger, so that the boarding cell 2 is not always blocked, bringing the boarding frequency to 2 s/person.

### 3.2. Existing Model Simulation

Nanchang Bus No. 245 currently uses the strategy of boarding at the front door and alighting at the back door. In this paper, simulations are carried out using Python programming software for different numbers of people getting off the bus (5, 10, 15, 20) to analyze the relationship between the number of boarders and the stopping time of the bus.

The simulation results show that when the number of passengers in the bus has no effect on the bus stopping time, there is an equilibrium point of 0.5 in the ratio of boarding to alighting, i.e., when the ratio of boarding to alighting exceeds 0.5, the stopping time is mainly related to the number of people boarding; conversely, it is related to the number of people alighting. The data show that when the number of alighting passengers is high, the alighting is not yet completed while the alighting passengers have already reached the vehicle, and there is crowding and conflict with the alighting passengers leading to a decrease in the alighting rate and thus a small increase in the stopping time (see Figure 7).

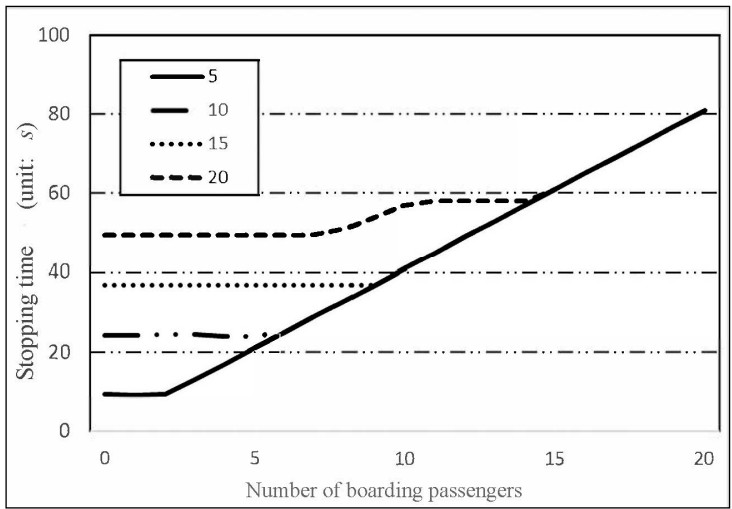

**Figure 7.** Diagram of the number of boarding passengers and bus dwell time.

*3.3. Comparison Strategies*

In this paper, four different bus boarding and alighting strategies are selected for comparison.

Strategy 1 considers the existing drop-off and pick-up strategy: get on at the front door, get off at the back door.

Strategy 2 supposes that after the disembarkation process is completed, one of the disembarkation cells can be transformed into an embarkation cell. The corresponding realistic scenario is that the rear door can be used to board the bus after all the disembarking passengers have left the bus, and the boarding passengers can board the bus at the rear door to improve the efficiency of passenger boarding and reduce the bus stopping time.

Strategy 3 suggests widening the drop door by 20 cm and then getting in and out of the car by a single side door.

Strategy 4 considers a combination of strategies 2 and 3; that is, when the lower doors are widened by 20 cm, the rear doors can allow boarding passengers to board at the rear doors after all exiting passengers have left the bus.

After the investigation of Bus No. 245, combined with the walking habits of pedestrians in real life, the safe interval needed for pedestrians to get off the stairs is 10 cm on each side, assuming that the shoulder width of each person is 50 cm. Due to the height difference between the disembarkation area and the ground, this means that the disembarkation door can satisfy two people at the same time when the door width is at least 130 cm. Therefore, strategy 2 can improve the disembarkation efficiency.

In this paper, simulations are carried out for different number of passengers in the vehicle, setting the number of boarders at 20 and the number of people getting off at 10. The simulation results show that strategy 4 can improve the operational efficiency by 40–50% at this boarding and alighting scale. Through the simulation, it is concluded that when the number of passengers in the vehicle is greater than 40 the passengers become slow to get on and off the bus due to the compressed safety space, and the bus operation efficiency decreases significantly, which corresponds to a full load rate of 65% (see Figure 8).

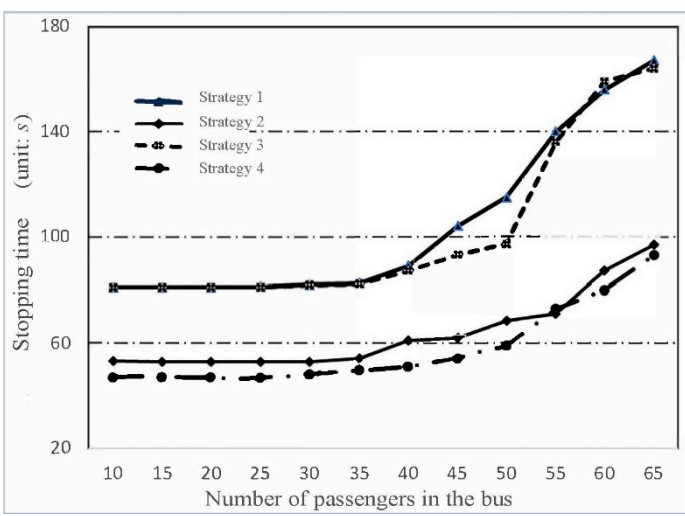

**Figure 8.** Simulation results of different strategies.

## 4. Conclusions

In this paper, complex pedestrian flows in and on buses were modeled, improving on the original dynamic parameter model. According to the CA theory, the potential energy is creatively used as the index of the direction parameter and adding attractiveness parameter, cell-occupancy parameter and forward parameter in order to simulate passenger interaction and passage behavior in dealing with congestion in a cell update. Its crowded traffic model can be applied to many complex scenarios, such as stations, lecture halls, etc.

Through simulation, it has been found that the bus stopping time at the station is related to the number of passengers inside the bus, the number of boarding passengers and the number of alighting passengers. When the number of passengers inside the bus is fixed, there is an equilibrium point for boarding and alighting passengers. When the ratio of boarding passengers to alighting passengers is greater than the equilibrium point, the bus stopping time is determined by the number of boarding passengers. Conversely, the bus stopping time is determined by the number of alighting passengers, and the passage of passengers in the bus affects the passenger passage rate, which in turn affects the bus passenger boarding and alighting rate. There is a full load rate point of 65%; when the passengers inside the bus are below this point, the passengers' boarding and alighting rates are uniform and the number of passengers inside the bus has no effect on the bus stopping time. When there are more passengers inside the bus than this point, the bus stopping time increases sharply. Therefore, this paper recommends that the bus full load rate is kept below 65% to ensure the efficiency of bus passenger boarding and alighting. If the bus full load rate is higher than 65%, it will increase the bus stopping time and seriously affect the bus operation efficiency; at this time it is recommended to replace the bus with one that has a larger capacity or to increase the frequency of departure. Through the simulation study of passengers getting on and off at the stops, it can provide feasible suggestions for the control strategy of the bus station.

By simulating and comparing the four bus boarding and alighting strategies, it is recommended that Bus No. 245, when newly procuring vehicles, selects buses with an effective door width of 130 cm at the rear door and operate in a mode where the rear door allows boarding after the completion of alighting, which can effectively reduce the bus stopping time at stops where the ratio of the number of boarding to alighting is greater than 0.5. The stopping time can be reduced by 40–50 percent at stations with a high number of boarders.

The improved CA adopts a hexagonal cell, which conforms to the general natural travel behavior characteristics of the human body, and the walking distance is the same in all directions. The application of potential energy difference to represent the direction parameter, although abstract but easy to understand, is the innovation of this paper. This

paper studies the full-load rate of the bus: in crowded conditions, two pedestrians generally pass through a cell after interaction. In some high-density cases, the space of a cell may be occupied by more than two passengers. In this case, the established model may fail.

In the study, due to the discrete nature of the CA model, the passenger boarding and moving speed is not accurate enough with the actual situation. When the bus stops, the relative position of the pedestrian and the bus stop will affect the bus boarding start time. In future research, the influence of platform settings and parking positions can be added. the simulation of passenger boarding and moving speed is not sufficiently accurate with the actual situation. In future research, the influence between the bus stops and the parking position can be added. In addition, the different ways of purchasing tickets for different passengers will also affect the time it takes for passengers to board the bus. These issues need to be discussed further.

In the next research study, it could also be possible to evaluate the model by comparing it to simulated situations with real people and different buses (or spaces simulating a bus) and different rules of entry–exit. Alternatively, considering the time of getting on and off the bus, the full load rate of the bus and the width of the door at the same time, a variety of simulation models can be used to simulate and analyze the bus passengers boarding and alighting to verify whether the CA model simulation is optimal. At the same time, we can apply the macroscopic passenger getting on and off characteristics obtained by station simulation to the control and management of bus platforms, such as scheduling vehicles according to the bus stop time or controlling the headway to avoid the bus-bunching phenomenon. On the other hand, we can introduce advanced technologies such as telematics and driverless into bus operation management, and can simulate bus operation scenarios in a telematics environment. The reference paper [36] uses an integrated Fuzzy Analytical Hierarchy Process (F-AHP) and Fuzzy Measurement Alternatives and Ranking According to Compromised Solutions (F-MARCOS) method to study the air-service quality, and in the future research we can consider similar methods to evaluate the bus service quality of different boarding strategies proposed in this paper.

**Author Contributions:** The individual contribution and responsibilities of the authors are listed as follows: Y.X. designed the research and developed the model; M.Z. analyzed the data and wrote the manuscript; L.X. collected survey data, constructed the model, and conducted model validation; B.Z. provided some comments and helped edit the manuscript; H.T., C.T. and Q.K. edited the manuscript; H.G. provided some comments on the case study and edited the manuscript. All authors have read and agreed to the published version of the manuscript.

**Funding:** This research was funded by the National Natural Science Foundation of China (Grant No. 71961006, 71971005, 52162042), Key Project of Jiangxi Provincial Social Science Foundation (21YJ03) and Graduate Student Innovation Special Fund Project of Jiangxi Province, China (YC2021-S456).

**Informed Consent Statement:** Informed consent was obtained from all subjects involved in the study.

**Data Availability Statement:** The survey data used to support the findings of this study are available from the corresponding author upon request.

**Acknowledgments:** The authors give thanks to the Nanchang Public Traffic Company for operational data. The authors are also very grateful for the comments from the editor and the anonymous reviewers.

**Conflicts of Interest:** The authors declare no conflict of interest.

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
