# Peer review of "Simulation Analysis of Bus Passenger Boarding and Alighting Behavior Based on Cellular Automata"

_sustainability, doi:10.3390/su14042429_

Round 1
Reviewer 1 Report
I had a problem with the validation of the model proposed and the empirical basis of it. Consequently I asked the authors to develop the references on the real behavior of bus passengers that should be the basis of their model, or to develop the "survey data" that they mention in the text. They did not complete any of these requirements. Consequently this model makes several hypotheses on the way passengers behave when passengers get into a bus and find a place, but these hypotheses are not founded (on the litterature or on a filed study). It seems that the authors do not understand this critical point. Can we build models of psycho-social behaviors just from personal intuitions? is this way of proceeding scientific? I really have a big problem with the validity of this procedure and it is not sufficient that the authors add in the very end: "It the next research it could also be possible to evaluate the modeling by comparing...".
Reviewer 2 Report
The paper has improved from the previous version.
Minor changes:
Review the conclusions, some ideas are repeated.
From my point of view, it would be interesting to study the relationship between the proposed strategies and the bottleneck reduction or efficiency of the urban transportation network; or how to monitor these strategies for sustainable development. (future research?)
Reviewer 3 Report
Review report for the paper “Simulation analysis of bus passenger boarding and alighting behavior based on cellular automata”.
Significance :
. The scientific content of this paper is correct.
. The results could be better presented. This would emphasize the quality of the presented work.
. The technical quality of this paper is correct.
. The conclusion is correctly justified and but it should be better supported by the results.
. The limits of the results obtained in this paper are not mentioned. This point should be investigated.
Quality of presentation:
. The abstract is clear and presents correctly the subject addressed in this paper.
. Introduction - What are your contributions?
. Better highlight novelty in the study.
. Better define motivations for the research.
. The data and analyses should be better presented. Add more discussion on the results. Add comparisons with existing approaches.
. Literature review is missing. Based on LR you should define gap you are trying to cover. As a part of that I suggest author read and add below interesting papers:
Application of Fuzzy AHP and Fuzzy MARCOS Approach for the Evaluation of E-Service Quality in the Airline Industry. Decision Making: Applications in Management and Engineering, 4(1), 127-152. https://doi.org/10.31181/dmame2104127b
. The conclusion section seems to rush to the end. The authors will have to demonstrate the impact and insights of the research. The authors need to clearly provide several solid future research directions. Clearly state your unique research contributions in the conclusion section. Add limitations of the model.
Scientific soundness :
. The subject addressed in this paper is relevant.
Interest to the readers :
. In my opinion, method of this paper seem to be interesting for the readership of the journal.
Author Response
Please see the attachment.

This manuscript is a resubmission of an earlier submission. The following is a list of the peer review reports and author responses from that submission.
Round 1
Reviewer 1 Report
The paper simulates passenger flows inside a bus based on a cellular automata model. The objective is to improve bus operation efficiency at bus stops. The model is carried out in bus number 245 of Nanchang, China. The main result shows that when the passengers loading rate in the bus reaches 65%, the bus stopping time will increase. Furthermore, some strategies related to door width are also explained.
From my point of view, some points could be improved:
Title: It would be interesting to add some reference that the study focuses on passengers’ flow inside the bus, that is, on-board passengers. At a first glance, it seems that the paper focuses on passengers at bus stops.
Abstract: The abstract is too long. I recommend summarizing the conclusions because they take up half of the paragraph.
There is room for improvement for Section 1 because the research gap to cover is not clear. What is the problem? Do not exist previous papers in this field? Not only related to the models, but papers about passenger flow inside a bus.
The state-of-the-art is too brief and not clear. It is not sufficiently clear the differences, advantages, and disadvantages among the cited models. Even, the section could be divided into two sections. These suggestions will help to gain more visibility in the scientific community.
What about sustainability? Some general statements about the concept but there are no clear examples.
Where is the methodology? How the model can be applied to other examples? There are some items, but it is not enough to understand the process. Are there some limitations? What about the capacity of the buses? Where the characteristics of the bus are indicated in the process?
In detail, what is the potential energy? It is presented on page 4 but it is not explained until section 2.2 (page 6). Could you improve the explanation of this concept? I understand it as a binary variable, doesn’t it?
I would appreciate further explanations in relation to equations 5 and 6 and the interaction judgment.
On page 7, section 2.3, the text says: “Pedestrians have 7 optional locations as their next target location.” Should it not be 6 optional locations?
Some strategies consider different sizes of the drop door. Is this realistic to implement? How much will be the investment to buy new buses?
In section 3.3 I recommend that the strategies consider/suppose instead of “to be/strategy is…” (writing format).
Finally, conclusions should be improved based on previous comments. What about the drawbacks of the proposed model? What about future research?
Reviewer 2 Report
In this article, the authors try to model complex passengers flows, boarding and alighting inside a bus, in order to compare 4 organizations of the space (varying width of doors and authorized entries and exits), in order to reduce bus stopping time. The issue is interesting and important if we want to encourage travelers to use public transport and to reduce autosolism.
The paper is mainly based on the comparison of 4 models of pedestrians flow simulation and on following development of one of these models, the “cellular automaton model”. It defines the parameters of the model (e.g. a seat is more attractive than a standing place, each one corresponding to a cell), then gives the calculation method of these parameters.
I am not a specialist of these models and cannot evaluate if the choice of this model instead of another is justified, and if its explanation is exhaustive enough, but I have a problem with the fact that there is no reference to studies of the behaviors of the passengers in real situations. The authors say “according to the investigation of public transport” but there is no references and no more details about that. They also say “the survey data yielded that the average ticketing time…” but we have no more information about this survey. So the references to the effective behaviors of the passengers is lacking. However, they should be a basis for the model. Either the authors should develop how they obtained these data or they must refer seriously to the literature in the domain, they are lacking by now.
It could also be possible to evaluate the model by comparing it to simulated situations with real people and different buses (or spaces simulating a bus) and different rules of entry-exit. That would give a real validity to their model.
N.B.: The English form is understandable but it could be improved by a native speaker.
Reviewer 3 Report
This study proposes a simulation analysis framework to identify boarding and aligning behavior in a bus based on cellular automation approach. It seems to be interested and have a merit on public transport field. However, needs to get more elaborate manuscript for publication in this journal by following few items below.
- A thorough literature review should be conducted. I am not following which contributions on this manuscript compared to methodologies for simulating analysis of boarding and aligning behavior inside a bus. Please clarify this.
- Second, please make more clear explanation, read fairly not well on each step and definitions. Schematic concepts give easily understood but all equations are not stranded. Try again get readable more.
- Few typo and different size of words are found.
Round 2
Reviewer 1 Report
I agree with almost all responses from the authors. However, from my point of view, question 6 about the methodology is not answered correctly.
It is needed a coherent and logical scheme (step by step) that guides other researchers to apply the described method to other examples, that is, a systematic way to solve the problem. I understand the research methods but following the process is not easy.
In fact, with a good methodology, their comments in the second bullet (the aspects to modify in other examples) should appear in the methodology. The work plan of research.
Minor changes in question 11: To correct the sentences that are grammatically incorrect (subject – verb agreement).
Reviewer 2 Report
The responses are not very satisfying (we asked for the methodology of the survey, and the importance of an evaluation of the model does not seem understood) and the paper is very model-oriented, we lose sight of sustainability in my opinion. That's the reasons why I would recommend submitting their work to another journal.